# The Correlation between Neck Pain and Disability, Forward Head Posture, and Hyperkyphosis with Opium Smoking: A Cross-Sectional Study from Iran

**DOI:** 10.3390/brainsci13091281

**Published:** 2023-09-03

**Authors:** Omid Massah, Amir Masoud Arab, Ali Farhoudian, Mehdi Noroozi, Fahimeh Hashemirad

**Affiliations:** 1Substance Abuse and Dependence Researcher Center, University of Social Welfare and Rehabilitation Sciences, Tehran 1985713834, Iran; noroozimehdi04@gmail.com; 2Department of Physiotherapy, University of Social Welfare and Rehabilitation Sciences, Tehran 1985713834, Iran; arabloo_masoud@hotmail.com (A.M.A.); fhashemirad@yahoo.com (F.H.); 3Department of Psychiatry, Tehran University of Medical Sciences, Tehran 1461884513, Iran; farhoudian@yahoo.com; 4Department of Psychiatry, University of Social Welfare and Rehabilitation Sciences, Tehran 1985713834, Iran; 5Psychosis Research Center, University of Social Welfare and Rehabilitation Sciences, Tehran 1985713834, Iran

**Keywords:** opiates, opium smoking, neck pain, neck disability, forward head posture, hyperkyphosis, drug use disorder, Iran

## Abstract

Opium smoking has been a common practice in Iran for many years, with people often smoking for long hours. During the COVID-19 pandemic, there was an increase in opium smoking due to false beliefs about its protective effects against COVID-19 infection. In this study, we aimed to examine the association between the non-ergonomic positions associated with traditional opium smoking in Iran and the development of neck pain and disability, forward head posture (FHP), and hyperkyphosis (HK). In this cross-sectional, correlational study, a total of 120 individuals who smoked opium were selected based on the inclusion criteria. They were interviewed about their addiction profile using the Lite version of the Addiction Severity Index and the Leeds Dependence Questionnaire. The presence of neck pain and disability was also evaluated using the Visual Analog Scale and the Neck Disability Index. The participants were examined for FHP via side-view photography and for HK using a flexible ruler. Data were analyzed using correlation coefficient tests and stepwise linear regression analysis. Based on the results, homelessness, the lifetime duration of opium smoking (in months), the duration of daily opium smoking (in minutes), and the severity of drug dependence had significant relationships with the severity of neck pain, neck disability, FHP, and HK. Homelessness was the strongest predictor of neck pain and disability (R^2^ = 0.367, *p* < 0.001), FHP (R^2^ = 0.457, *p* < 0.001), and HK (R^2^ = 0.476, *p* < 0.001), followed by the lifetime duration of opium smoking and the duration of daily opium smoking, respectively, in which R^2^ increased to 0.505 (*p =* 0.011), 0.546 (*p =* 0.022), and 0.570 (*p =* 0.004) with the addition of two other variables. Overall, an increase in the duration of sitting in non-ergonomic positions could lead to neck pain and disability, FHP, and HK due to the non-neutral posture of opium smokers.

## 1. Introduction

Most researchers and scientists agree that addiction is a brain disease with a neurobiological basis. While there may be some disagreement on the specific neurological symptoms, it is widely accepted that addiction leads to both substance dependence and addictive behaviors. In other words, in addition to developing a dependence on the consumption of a substance, an individual may also develop a behavioral addiction to the method and tools used to consume that substance [1,2].

Iran is one of the countries that faces significant challenges in the area of addiction. According to the 2022 World Drug Report, Iran has seized more opium, morphine, and heroin than any other country around the world. According to this report, 98% of the opium seized worldwide comes from Iran. This is due to the country’s position as an international transit route, as well as its long history and culture of opium smoking, which is tied to the region’s geopolitics [3,4]. According to statistics, more than 2,000,000 people are using drugs daily in Iran, and more than 70% of them are opiate users [5]. 

Many addictive behaviors have been shown to have a similar pathogenesis to substance use disorders involving the same neurocircuitry [6], and opium smokers seem to have both addictive behaviors and substance dependence. In other words, smoking opium is an addictive behavior that can compound an individual’s dependence on the substance itself. Opium smoking is prevalent in Iran due to its historical background and unique geographical conditions. Unfortunately, the outbreak of the COVID-19 pandemic and false beliefs about the protective effects of opium against the virus have led to an increase in opium smoking [7]. It is worth mentioning that more than 90% of drug use in Iran is through smoking [5]. 

In Iran, drug use disorders are ranked fourth among health problems that cause the most disabilities. Also, in terms of factors that contribute to morbidity and mortality, drug use disorders rank eighth [8]. Drug users seem to be more prone to some musculoskeletal disorders for various reasons, such as non-ergonomic positions when smoking or inhaling, lack of movement and inactivity [9], malnutrition [10], and heavy smoking [11]. When opium dependents smoke opium, they use three special tools (Figure 1) called ‘Vafour’ (a device for smoking opium, including a wooden pipe and a small clay or ceramic pot attached to the pipe), ‘Gholgholi’ (a small water pipe or hookah), and ‘Sikh-o-Sang’ (a method of opium smoking where a hot and melted spoke is touched to a piece of opium which is placed on a small metal pin) in uncomfortable positions where their head tilts forward more than normal. This can also cause changes in their spine, resulting in a curved posture and sometimes leaning to one side [12,13]. If opium smoking in the common non-ergonomic position continues for several hours a day over many years, the risk of developing neck musculoskeletal disorders will be high. This is similar to many work-related musculoskeletal disorders caused by non-ergonomic positions [14,15].

Musculoskeletal disorders are among the most common health concerns worldwide. In the 2017 classification of disabling health problems, musculoskeletal disorders were among the leading causes of disability and reduction in years lived without disability. Lower back and neck pain ranked fourth among these problems [16,17]. Also, according to a similar report from 2020, spinal problems and headaches were among the top 10 causes of disability in the age groups of 10–24 and 25–49 years [18]. Musculoskeletal disorders are often acquired and caused by work or non-ergonomic positions [19]. Various physical and mechanical risk factors, such as long-term repetitive work (especially in a non-ergonomic position), continuous lifting or lifting of heavy loads, and pushing, pulling, or carrying heavy loads, can cause or aggravate these disorders. It has been shown that there is a close relationship between these disorders and improper use of body mechanics [20,21]. 

Bending forward and turning the neck, incorrect sitting and standing positions, and repetitive manual activities are among the most significant factors contributing to neck musculoskeletal disorders. There is also a positive relationship between neck pain and incorrect sitting or standing postures. Moreover, neck and shoulder muscle and joint problems are significantly related to poor posture [15,22]. Generally, neck pain and neck disability are among the most common work-related musculoskeletal disorders [23]. They are common in many occupations, as well as in repetitive daily activities and even sports. These conditions can arise when the upper body and upper limbs are held in uncomfortable positions or when the upper limb is repeatedly held up and forward [24,25]. 

While smoking opium, individuals often hold their upper limbs and hands up and forward, and this position is repeated for several hours a day. Forward head posture (FHP) and hyperkyphosis (HK) are also very common among individuals who engage in daily activities, sports, and jobs that involve non-ergonomic positions. For instance, office workers, industrial brokers, and even some athletes, such as cyclists, show higher rates of FHP and HK [26,27,28,29,30]. 

When opium smokers sit and smoke opium, they often tilt their head and neck forward, creating a kyphotic hump in their spine. In fact, opium smoking by traditional methods in Iran is performed in a completely non-ergonomic position, and it is strange that it has not been given serious attention until now. Given the increase in physical disorders in today’s urban life, it is important to document the incidence and prevalence of these disorders, as well as the underlying factors, across different age groups, genders, occupations, and other demographic factors. After many years of simultaneous research and treatment in substance use disorder, and many clients who complained of postural deformities and pain and problems caused by it, and due to limited and inconclusive research conducted on posture deformities due to drug use disorder and drug use-related musculoskeletal disorders, both in Iran and other countries, and the fact that addiction research has mainly focused on psychiatric, psychological, and social aspects of this phenomenon, in this study, we aimed to investigate the relationship between opium smoking and the incidence of neck pain and disability, FHP, and HK among opium smokers in Tehran, Iran.

## 2. Materials and Methods

### 2.1. Participants

This cross-sectional, correlational study was conducted in 2022 in Tehran, Iran. Through purposive sampling, a total of 120 opiate users were selected among the clients of outpatient treatment centers in Tehran based on the inclusion criteria and were evaluated for the presence of neck pain and disability, forward head posture, and hyperkyphosis. The primary criterion for inclusion in this study was a diagnosis of substance use disorder and dependence, as defined by the International Classification of Diseases (ICD-11) criteria [31]. The participants’ primary drug of choice was opium, which was predominantly consumed via smoking. Other inclusion criteria were the ability to stand, being in the age range of 25–50 years, and having a body mass index (BMI) below 27.5 kg/m^2^. These criteria were selected because individuals in this age range, with a BMI below 27.5 kg/m^2^, are less likely to have musculoskeletal disorders [32,33,34,35].

On the other hand, the exclusion criteria were a history of neuromuscular or skeletal disorders, a history of surgery in the spine and shoulder girdle, a history of championship or regular exercise, any imbalance caused by a specific disease, any clear postural deformity or anatomical disorder, and use of a smartphone or a tablet for more than 0.5–1 h per day [36]. The sample size for this study was determined using G*Power software, based on the number of variables in cross-sectional correlational studies, with a statistical power of 80% and a significance level of 0.05. The final sample size was estimated to be 105 participants.

### 2.2. Tools and Data Gathering

The data collection tools used in this study included the following: 1. A demographic questionnaire. 2. The Lite version of the Addiction Severity Index: ASI-lite is a tool for the assessment of different aspects of drug use where the addiction history and drug use section, in particular, was used in this study in order to collect data on the history of drug use [37]. 3. The Persian version of the Leeds Dependence Questionnaire: a ten-item tool that measures the severity of substance dependence from a psychological aspect and is scored on a Likert scale of zero to three, and its validity and reliability have been confirmed [38]. 4. The Visual Analog Scale (VAS), which is a simple, valid, and reliable tool for pain measurement [39]. 5. The Neck Disability Index (NDI): a ten-item scale that assesses the impact of neck pain and disability in daily life tasks. However, each item is scored from zero to five [40]. Measurements were also obtained using photography and a curved ruler, which are valid and reliable methods for evaluating FHP and HK [41,42]. To evaluate FHP, the craniovertebral angle was measured. For the measurement of the craniovertebral angle, a side-view photograph was taken. For this purpose, the person was photographed from a distance of 265 cm using a Canon digital camera (Canon PowerShot G11 10 MP, Japan). The participant stood sideways next to a wall, and the camera was positioned at shoulder height and perpendicular to the sagittal plane of the participant’s body. Finally, the photos were transferred to AutoCad 2013 software, and the craniovertebral angle was calculated [41]. 

Subsequently, the examination of the spine curvature in the thoracic region and the measurement of the thoracic kyphosis angle were performed. For this purpose, the participants stood in a normal and relaxed position. They then moved their head, neck, and upper limbs several times to release any muscle tension or contraction. Afterward, the participants stood upright in a motionless posture with their spines in a completely neutral, straightened position. A flexible ruler (24-inch, Staedtler Mars, Nuernberg, Germany) was placed tangentially on the thoracic spine, spanning from T2 to T12. The curve of the ruler was then transferred to a sheet of A3 paper by carefully maintaining its shape. A curved line was finally drawn on the paper to represent the curvature of the thoracic spine. The kyphosis angle was measured using the following equation [42]: α = 4 arctan (2 H/L).

### 2.3. Data Analysis

Statistical analysis was performed in SPSS Version 23, using the Shapiro–Wilk test, correlation coefficient tests (Goodman and Kruskal’s lambda, Pearson’s correlation coefficient test, and Spearman’s correlation test), and stepwise linear regression analysis at a significance level of <0.05.

### 2.4. Ethical Consideration

The study design and research method were approved by the ethics committee of the University of Social Welfare and Rehabilitation Sciences with the code of IR.USWR.REC.1398.120. We obtained informed consent from all participants. This article is extracted from the doctoral thesis of the first author.

## 3. Results

### 3.1. Basic Variables

The demographic characteristics of the samples and their substance use profile are presented in Table 1.

### 3.2. Main Variables

There were significant relationships between the severity of neck pain and the age of continuous opium smoking, homelessness, the lifetime duration of opium smoking (in months), and the duration of daily opium smoking (in minutes). However, the age of occasional opium smoking, BMI, occupation, drug use duration, opium smoking method, and drug dependence severity had no significant relationships with neck pain (Table 2). Meanwhile, neck disability showed significant correlations with homelessness, the lifetime duration of opium smoking (in months), the duration of daily opium smoking (in minutes), and drug dependence severity. On the other hand, there was no significant relationship between neck disability and BMI, age of occasional or continuous opium smoking, and drug use duration (Table 3).

To investigate the correlation of FHP and HK with independent variables, the correlation coefficients were calculated. According to the results presented in Table 4, FHP and HK had significant moderate reverse correlations with the age of continuous opium smoking. Homelessness had the strongest correlation with FHP and HK. Also, the lifetime duration of opium smoking (in months) and daily opium smoking (in minutes) were strongly correlated with FHP and HK. Moreover, drug dependence severity had a significant moderate correlation with FHP and HK.

To assess the impact of the independent variables on the occurrence of neck problems and disabilities, FHP, and HK, a stepwise regression analysis was performed. As presented in Table 5, Table 6 and Table 7, homelessness was the strongest predictor of neck disability (R^2^ = 0.367, *p* < 0.001), as well as FHP (R^2^ = 0.457, *p* < 0.001), and HK (R^2^ = 0.476, *p* < 0.001). Also, the lifetime duration of opium smoking and the daily duration of smoking were other variables that increased the coefficient of determination (R^2^ and adjusted R^2^) in the regression analysis for predicting neck disability and HK. R^2^ increased during two stages after entering two other variables into the regression equation. In fact, R^2^ increased to 0.456 (*p* = 0.004) and 0.505 (*p =* 0.011) for neck disability and 0.528 (*p* = 0.002) and 0.570 (*p =* 0.004) for HK. However, in the study of FHP, it was found that the daily duration of opium smoking was a stronger predictor than the lifetime duration of opium smoking in the regression analysis, and R2 increased to 0.504 (*p* = 0.015) and 0.546 (*p =* 0.022), respectively (Table 5, Table 6 and Table 7).

## 4. Discussion

In this study, we aimed to investigate the correlation of opium dependence, its severity, and opium smoking with neck pain and disability. The present findings revealed that an increase in opium smoking can be related to negative effects on neck function, and it is also correlated with neck pain and limitation. Evidently, a sedentary lifestyle, which is common in today’s modern societies, is a source of many health concerns [43], and Iran occupies a unique position among other countries in the world due to the reasons mentioned earlier. In Iran, it is common for opium addicts to smoke opium (or residues of smoked opium) for several years and for long hours a day using various tools, such as ‘Vafour’ and hookah, in non-ergonomic positions [12].

Opium use can be a major risk factor for the development of postural disorders and skeletal problems in the upper body. Opium smoking for long hours is in contrast to the shorter smoking time associated with heroin and cannabis use, which are more common in other countries [3]; therefore, this issue may not be as important in other countries as it is in Iran. Currently, there are no similar studies from other countries that can be compared with the results of our study. The only established finding of previous research is the impact of homelessness on the occurrence of health problems. For instance, in a study by Sun et al., a significant direct correlation was found between homelessness and increased pain and musculoskeletal disorders [44]. This relationship has also been confirmed in other previous research [45].

Iran has the second highest rate of severe opioid addiction in the world, with the highest per capita rates of heroin and opium addiction. Among the population aged 15–60 years, one in every five individuals engages in non-permanent drug abuse, while one in every seventeen individuals is a permanent user [3]. The non-ergonomic and harmful posture that drug users adopt while smoking opium can cause pain and disability in the cervical vertebrae. Moreover, the results of our study found a significant relationship between the occurrence of neck pain and the younger age of continuous drug use, the method of use, the lifetime duration of opium smoking, and the duration of daily opium smoking. However, there is no similar study in the literature to compare the present results with other countries. In this regard, only a study by Daneshmandi et al. showed that spine disorders and posture deformities were more prevalent among prison addicts in Iran [46]. To explain the occurrence of pain, it can be claimed that repetitive work in an incorrect and non-neutral position can lead to postural pressure, fatigue, and pain [47]. It has also been shown that people often adopt incorrect and non-standard body positions due to their work habits or job requirements. Over time, these positions can cause various types of pain, known as postural pain [48]. 

Similar to the findings of the present study on opium smokers, previous studies have also found that dentists who work for several hours a day in harmful positions are at risk of developing various types of neck musculoskeletal disorders [49,50]. Prolonged sitting in non-ergonomic positions has a proven correlation with neck pain. As in Rahmani’s study, this has been proven about dentists [51]. Additionally, this has been shown in employees who work with computers and keyboards [52].

In this study, FHP had a significant correlation with the lifetime duration of opium smoking and the daily duration of opium smoking, while it was not correlated with the total duration of drug use (any type of substance and any method of use). These results highlight the relationship between opium smoking and FHP and suggest a causal relationship between these two variables. Although there may not be any similar studies on this issue, we can refer to other studies that are somewhat similar to our study. For instance, the prevalence of FHP among office workers using desktop computers and gamers who often adopt forward and non-neutral head and neck positions for extended periods of time is similar to that observed among opium smokers [53,54]. 

Although the relationship between opium smoking and forward head posture has not been reported in any study, some studies have shown a correlation between forward head posture and addiction [46,55]. Also, the strong correlation between neck pain and forward head posture [56] can be a sign of the need for more studies on opium smokers.

Additionally, HK had a significant positive correlation with the lifetime duration of opium smoking and the daily duration of opium smoking. This correlation should be investigated more thoroughly in future studies to determine if there is a causal relationship. If it is confirmed that dentists and farmers adopt similar postures during their work, it can be suggested that they are at an increased risk of developing HK in the long term [57,58]. Also, in another study, neck disability, which was significantly correlated with opiate smoking in our study, was more common among office workers who were in similar non-ergonomic positions [59].

In this study, according to the regression analysis, homelessness was the strongest predictor of musculoskeletal disorders in the neck. Although no similar research has been conducted to support the significant relationship observed in this study, we can suggest secondary causes related to homelessness that may explain this relationship. For instance, homelessness can lead to a lack of physical activity, unhealthy and insufficient nutrition, non-ergonomic sitting and sleeping positions, and heavy smoking, all of which can exacerbate the occurrence of skeletal disorders. 

Previous studies have shown that neck pain and musculoskeletal disorders have a positive correlation with incorrect sitting, standing, or sleeping positions [22,58,60]. Research also suggests the effect of malnutrition on muscle weakness, especially in the axial muscles, which can cause disability and spinal deviation [61]. Also, studies have shown that smoking can lead to tissue malnutrition by causing capillary disorders, which can result in musculoskeletal pain in individuals who lift heavy loads for their jobs [62]. Additionally, the occurrence of pain and musculoskeletal disorders caused by hypovitaminosis D associated with malnutrition has been previously approved [63].

Some limitations in this study need to be addressed. First, the primary limitation related to this topic (not this specific study) is the lack of similar research both in Iran and other countries. In fact, after considering the possibility of complications arising from non-ergonomic positions in opium smokers in Iran, a review of the literature revealed no similar reports. It appears that this issue may not be as significant in other countries, perhaps due to shorter smoking durations or differences in the type of drugs consumed. Similarly, this topic has not been discussed yet in Iran. Therefore, it was necessary to design our research using more basic methods to establish the existence of significant correlations. It is recommended to use research methods, such as case–control and prospective studies, to strongly establish the correlations found in this study. The second limitation of our study is that female participants declined to participate, and we had to rely solely on the results of the male participants. Another limitation of this study is the possibility of less accurate answers, underestimation, and even denial in some answers. Therefore, it is better to pay attention to these issues in future studies.

## 5. Conclusions

Non-ergonomic positions due to opium smoking have a strong relationship with neck pain and disability and posture deformities such as forward head posture and hyperkyphosis. Opium smoking through different methods for long hours a day, which continues for years, can lead to neck pain, neck musculoskeletal disorders, a decrease in craniovertebral angle (forward head posture), and an increase in thoracic kyphosis angle (hyperkyphosis).

## Figures and Tables

**Figure 1 brainsci-13-01281-f001:**
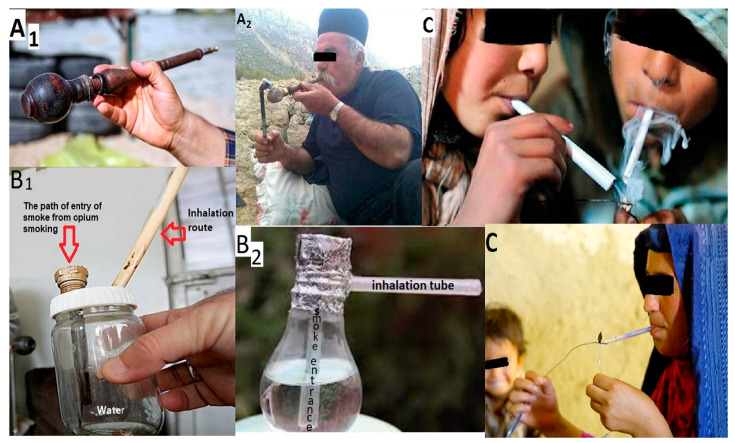
Different types of tools used for smoking opium in Iran: (**A_1_**) Vafour, (**A_2_**) opium smoking with a Vafour; (**B_1_**) a type of Gholgholi, (**B_2_**) another type of Gholgholi; (**C**) Sikh-o-Sang.

**Table 1 brainsci-13-01281-t001:** Demographic characteristics and opium use profile of participants.

Variables	Mean	SD
Age (Year)	39.30	5.05
Weight (Kg)	72.70	6.48
Height (m)	1.73	6.13
BMI (Kg/m^2^)	24.29	2.12
Age of first drug use (any kind of drug)	19.35	6.45
Age of occasional drug use (any kind of drug)	21.05	5.15
Age of occasional opium smoking	23.45	5.85
Age of continual opium smoking	28.20	6.15
Duration of opium smoking during life (months)	110.45	31.70
Duration of daily opium smoking (minutes)	212.35	48.45
Dependence severity score	25.20	4.35

BMI—body mass index.

**Table 2 brainsci-13-01281-t002:** The results of correlation coefficients between neck pain and predictor variables studied in opium smokers.

Variables		Correlation Coefficient	*p*-Value
Predictor Variable	Criterion Variable		
Age of occasional opium smoking	VAS	−0.19	0.112
Age of continual opium smoking	VAS	−0.23	0.040
BMI	VAS	0.29	0.077
Job	VAS	0.22	0.096
Homelessness	VAS	0.61	˂0.001
Opium smoking method (based smoking tool)	VAS	0.24	0.135
Drug use duration	VAS	0.36	0.080
Duration of opium smoking during life (months)	VAS	0.53	˂0.001
Duration of daily opium smoking (minutes)	VAS	0.59	0.022
Drug dependence severity	VAS	0.39	0.072

BMI—body mass index; VAS—Visual Analog Scale.

**Table 3 brainsci-13-01281-t003:** The results of correlation coefficients between total score of neck disability index and predictor variables studied in opium smokers.

Variables		Correlation Coefficient	*p*-Value
Predictor Variable	Criterion Variable		
Age of occasional opium smoking	NDI	−0.29	0.090
Age of continual opium smoking	NDI	−0.46	0.065
BMI	NDI	0.25	0.089
Job	NDI	0.38	0.113
Homelessness	NDI	0.69	˂0.001
Opium smoking method (based smoking tool)	NDI	0.41	0.045
Drug use duration	NDI	0.38	0.105
Duration of opium smoking during life (months)	NDI	0.61	0.007
Duration of daily opium smoking (minutes)	NDI	0.68	0.005
Drug dependence severity	NDI	0.48	0.04

BMI—body mass index; NDI—Neck Disability Index.

**Table 4 brainsci-13-01281-t004:** Correlation of forward head posture (FHP) and hyperkyphosis (HK) with predictor variables studied in opium smokers.

Variables		Correlation Coefficient	*p*-Value
Predictor Variable	Criterion Variable		
Age of occasional opium smoking	FHP	−0.29	0.079
	HKP	−0.26	0.82
Age of continual opium smoking	FHP	−0.56	0.045
	HKP	−0.49	0.022
BMI	FHP	0.35	0.069
	HKP	0.33	0.061
Job	FHP	0.29	0.081
	HKP	0.31	0.105
Homelessness	FHP	0.71	˂0.001
	HKP	0.77	˂0.001
Opium smoking method (based smoking tool)	FHP	0.37	0.115
	HKP	0.49	0.066
Drug use duration	FHP	0.38	0.105
	HKP	0.59	0.033
Duration of opium smoking during life (months)	FHP	0.66	0.005
	HKP	0.71	˂0.001
Duration of daily opium smoking (minutes)	FHP	0.65	0.002
	HKP	0.74	˂0.001
Drug dependence severity	FHP	0.51	0.03
	HKP	0.59	0.007

BMI—body mass index; FHP—forward head posture; HK—hyperkyphosis.

**Table 5 brainsci-13-01281-t005:** Stepwise regression analysis of neck disability (criterion variable) based on the predictor variables studied in opium smokers.

Step	Predictor Variable	R	R^2^	Adjusted R^2^	β	t	*p*-Value
1	Homelessness	0.605	0.367	0.309	0.521	4.254	˂0.001
2	Duration of opium smoking during life (months)	0.675	0.456	0.407	0.401	3.140	0.004
3	Duration of daily opium smoking (minutes)	0.711	0.505	0.460	0.345	2.405	0.011

**Table 6 brainsci-13-01281-t006:** Stepwise regression analysis of forward head posture (FHP) as the criterion variable based on predictor variables studied in opium smokers.

Step	Predictor Variable	R	R^2^	Adjusted R^2^	β	t	*p*-Value
1	Homelessness	0.676	0.457	0.443	0.539	4.414	˂0.001
2	Duration of daily opium smoking (minutes)	0.710	0.504	0.492	0.411	3.338	0.015
3	Duration of opium smoking during life (months)	0.739	0.546	0.534	0.382	2.720	0.022

**Table 7 brainsci-13-01281-t007:** Stepwise regression analysis of hyperkyphosis (HK) as the criterion variable based on predictor variables studied in opium smokers.

Step	Predictor Variable	R	R^2^	Adjusted R^2^	β	t	*p*-Value
1	Homelessness	0.690	0.476	0.463	0.572	4.705	˂0.001
2	Duration of opium smoking during life (months)	0.727	0.528	0.516	0.433	3.60	0.002
3	Duration of daily opium smoking (minutes)	0.755	0.570	0.559	0.386	2.925	0.004

## Data Availability

The raw data supporting the conclusions of this article will be made available by the authors without undue reservation.

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
