# Peer review of "The Correlation between Neck Pain and Disability, Forward Head Posture, and Hyperkyphosis with Opium Smoking: A Cross-Sectional Study from Iran"

_brainsci, 2023, doi:10.3390/brainsci13091281_

Round 1
Reviewer 1 Report
Please find in the following my comments about the review of a manuscript under the title (The Correlation between Neck Pain and Disability, Forward Head Posture and Hyperkyphosis with Opium Smoking in the Most Prevalent Opium Smoking Persian Gulf Country; a Cross-sectional Study from Iran). In this study, the authors aim to investigate the relationship between non-ergonomic positions of traditional opium smoking in Iran with the occurrence of neck pain and disability, forward head posture and hyperkyphosis.
Originality and relevance
· The study is interesting for reading.
· The study has mild scientific quality.
· The study is relevant to the scope of this journal.
· The manuscript is relevant to the field and its presentation needs major modifications to be clearer.
Comments:
Title:
· Make it concise. The suggested title: (The Correlation between Neck Pain and Disability, Forward Head Posture and Hyperkyphosis with Opium Smoking; a Cross-sectional Study from Iran
Abstract:
· Add more details about the methodology and mentioned the most prominent finding in the results.
· Need to be reorganized after consideration of the comments.
Introduction
· The rationale of the study is not clear in the introduction section.
Materials and methods
· Add references for all methods used.
· Add a reference for this sentence (age between 25 and 50 years, and Body Mass Index (BMI) below 27.5 (because in this age range and BMI, the probability of skeletal-muscular disorders is less)).
· In section 2.1. Data analysis: mention all the used statistics and identify the level of significance.
Results:
· The title and legends of the figures should be informative and self-explanatory? Revise.
· Define all the abbreviations mentioned in the tables.
· Results section is deficient and needs major modification to explain all the findings.
· Remove this section or modify it according to your result (Neck disability and disorders were almost twice as correlated with opium smoking through a ‘Wafour’ (a type of vape that is a traditional tool of opium smoking in the Middle East and Persian Gulf countries)).
Discussion:
· Discussion is deficient. The authors repeated the results in this section without any discussion or interpretation. The findings need more interpretation.
Language
· Extensive editing of English language and style required. There are many typos and grammatical errors. The punctuation should also be checked.
Extensive editing of English language required
Author Response
Title:
Make it concise. The suggested title: (The Correlation between Neck Pain and Disability, Forward Head Posture and Hyperkyphosis with Opium Smoking; a Cross-sectional Study from Iran
Reply: Was done.
Abstract:
Add more details about the methodology and mentioned the most prominent finding in the results.
Reply: Was done.
Introduction:
The rationale of the study is not clear in the introduction section.
Reply: It was more and better explained.
Materials and methods:
Add references for all methods used.
Reply: The references were added.
Add a reference for this sentence (age between 25 and 50 years, and Body Mass Index (BMI) below 27.5 (because in this age range and BMI, the probability of skeletal-muscular disorders is less).
Reply: Was done.
In section 2.1. Data analysis: mention all the used statistics and identify the level of significance.
Reply: Was done.
Results:
The title and legends of the figures should be informative and self-explanatory? Revise.
Reply: Was done.
Define all the abbreviations mentioned in the tables.
Reply: Was done.
Results section is deficient and needs major modification to explain all the findings.
Reply: The results were explained with more details.
Remove this section or modify it according to your result (Neck disability and disorders were almost twice as correlated with opium smoking through a ‘Wafour’ (a type of vape that is a traditional tool of opium smoking in the Middle East and Persian Gulf countries).
Reply: That section was removed. That section was related to other parts of thesis.
Discussion:
Discussion is deficient. The authors repeated the results in this section without any discussion or interpretation. The findings need more interpretation.
Reply: Discussion was developed with more interpretation. Of course, I have to emphasize and repeat that such a study on opium smokers and the relationship with opium smoking has never been done.
Language
Extensive editing of English language and style required. There are many typos and grammatical errors. The punctuation should also be checked.
Reply: The article was edited natively.
Reviewer 2 Report
The review of the manuscript entitled: “The Correlation between Neck Pain and Disability, Forward Head Posture and Hyperkyphosis with Opium Smoking in the Most Prevalent Opium Smoking Persian Gulf Country; a Cross-sectional Study from Iran”
The study aimed to analyze the correlations among daily and lifetime duration of traditional opium smoking and the occurrence of neck pain and disability, forward head posture and hyperkyphosis in Iran. For this purpose, 120 participants were recruited and authors used Maudsley addiction profile, Leeds dependence questionnaire, visual analog scale and neck disability index. The participants were also evaluated for forward head posture (FHP) through side view photography and hyperkyphosis (HK) applying a flexible ruler.
Comments for Authors:
Thank you for the valuable research you have done. However, there are some comments which I hope can help you to improve your paper:
1) ‘Methods’ section, ‘Tools and data gathering’ subsection: It is recommended that authors include a brief explanation about each questionnaire which is used and statistical properties of them.
2) Table 1: It is recommended that authors include the units of measurement for each variable.
3) ‘Discussion’ section, lines 215-217, the authors mentioned: “The findings of this study showed that increasing the opium smoking can have negative effects on some functions of the neck and cause pain and limitations in its ability”. However, this is a cross-sectional study and can prove ‘correlations’ only. Therefore, authors should be cautious about using statements which imply on ‘causality’.
4) It is recommended that authors include ‘denial or underestimation of amount and daily / lifetime duration of substance use’ of participants in the limitation paragraph of the manuscript.
5) ‘Data Availability Statement’ section should be edited.
6) It is recommended that authors include photos of 'Vafour', 'Gholgholi', and 'Sikh-o-sang'.
Good luck
The paper is overall understandable. However, it is full of writing and grammar errors and needs to be revised by a native.
Author Response
‘Methods’ section, ‘Tools and data gathering’ subsection: It is recommended that authors include a brief explanation about each questionnaire which is used and statistical properties of them.
Reply: Thank you for this recommendation. It was done.
Table 1: It is recommended that authors include the units of measurement for each variable.
Reply: Was done.
Discussion’ section, lines 215-217, the authors mentioned: “The findings of this study showed that increasing the opium smoking can have negative effects on some functions of the neck and cause pain and limitations in its ability”. However, this is a cross-sectional study and can prove ‘correlations’ only. Therefore, authors should be cautious about using statements which imply on ‘causality’.
Reply: Thank you for your attention. It was corrected.
It is recommended that authors include ‘denial or underestimation of amount and daily / lifetime duration of substance use’ of participants in the limitation paragraph of the manuscript.
Reply: Was added.
‘Data Availability Statement’ section should be edited.
It is recommended that authors include photos of 'Vafour', 'Gholgholi', and 'Sikh-o-sang'.
Reply: The photos were added.
Comments on the Quality of English Language
The paper is overall understandable. However, it is full of writing and grammar errors and needs to be revised by a native.
Reply: The manuscript was edited natively.
Round 2
Reviewer 1 Report
Please find in the following my comments about the review of a manuscript under the title (The Correlation between Neck Pain and Disability, Forward Head Posture and Hyperkyphosis with Opium Smoking; a Cross-sectional Study from Iran). In this study, the authors aim to investigate the relationship between non-ergonomic positions of traditional opium smoking in Iran with the occurrence of neck pain and disability, forward head posture and hyperkyphosis.
The authors revised their manuscript. I think the revisions have been completely conducted according to the reviewers' comments. There are no more comments required.